# Healthism *vis-à-vis* Vaccine Hesitancy: Insights from Parents Who Either Delay or Refuse Children's Vaccination in Portugal

Joana Mendonça *[ID] and Ana Patrícia Hilário [ID]

Instituto de Ciências Sociais, Universidade de Lisboa, Av. Prof. Aníbal Bettencourt 9, 1600-189 Lisboa, Portugal; patriciahilario@gmail.com
* Correspondence: joanamsmendonca@gmail.com

**Abstract:** Although healthism appears to be at the heart of the decision-making process of vaccine hesitancy, this matter has been understudied. We believe that the concept of healthism may be key to lessen the polarization of discourses around vaccination, offering a broad understanding of parents' decision to not vaccinate their children. This article aims to deepen the knowledge on the relation between healthism and vaccine hesitancy, using Portugal as a case study. A qualitative research approach was adopted, and therefore, in-depth interviews were conducted with 31 vaccine-hesitant parents. The findings showed that vaccine-hesitant parents usually adopt several strategies based on natural living to prevent and tackle their children's potential health issues. There appears to be a common approach towards health and life (i.e., healthism) among vaccine-hesitant parents. Drawing upon the healthism ideology, vaccine-hesitant parents make choices to ensure the good health of their child. These choices nevertheless represent a privileged position as the pursuit of healthfulness is constrained by sociodemographic aspects. Using vaccine hesitancy as the starting point, our findings show that healthism and its focus on personal accountability under the umbrella of neoliberalism may jeopardize global public health. Healthcare professionals should pay particular attention to healthism when addressing vaccine hesitancy in Portugal and elsewhere. Research evidence advocates the need to be sensitive to the broad spectrum of vaccine hesitancy as this encompasses multiple views on the subject.

**Keywords:** vaccine hesitancy; healthism; parenting; qualitative methods

## 1. Introduction

Vaccine hesitancy has been defined as one of the greatest threats to global health by the World Health Organization [1]. As a way of gaining a deeper understanding of this complex and context-specific phenomenon [2], social scientists have analyzed "how health decisions are formulated within social worlds" [3]. Nevertheless, very few studies to date have acknowledged the relation between healthism and vaccine hesitancy [4]. We argue that the phenomenon of vaccine hesitancy should be deeply examined in the light of the new health consciousness and movements as described by Crawford [5]. "Healthism is a well-recognized socio-cultural phenomenon in the western (and westernized) middle classes, characterized by high health awareness and expectations, information-seeking, self-reflection, high expectations, distrust of doctors and scientists, healthy and often 'alternative' lifestyle choices, and a tendency to explain illness in terms of folk models of invisible germ-like agents and malevolent science." [6]. Healthism draws from neoliberal discourse that highlights personal responsibility and the need for self-management. Within neoliberal society, individuals are held to be accountable for their own health behaviors and to manage health risks [7], meaning that they are responsible for their actions and decisions regarding their own health [8]. Following this train of thought, good health is only achievable through commitment and personal investment, requiring self-restraint and constant vigilance [9]. Healthism is deeply interwoven with individualism and accountability, which is at the heart

of neoliberal thinking [10]. We argue that vaccine hesitancy (e.g., the delay or refusal of vaccination) is explained by parents under the umbrella of healthism, in the sense that they believe that the maintenance of the good health of their children is a process that requires constant vigilance and is achievable through their personal investment and commitment. This article intends to increase the knowledge on the relation between healthism and vaccine hesitancy, using Portugal as a case study.

*The Context*

Portugal offers an interesting context to explore vaccine hesitancy. In Portugal, vaccination is not compulsory except for the tetanus and diphtheria vaccines. Vaccination is free of charge and preferably administered in public healthcare centers or in certain cases in private hospitals, even though parents can decide to immunize their children in private practices by only paying for the administration of the vaccine. Vaccination is not compulsory for enrolment in public schools in Portugal; however, the schools are obliged to inform the healthcare center of their neighborhood when a child is not vaccinated, as established in the National Vaccination Program [11]. Although Portuguese parents have been found to be the most confident in vaccines in comparison to parents from other European countries [12], two measles outbreaks occurred in the country in 2017 [13]. This shows that vaccine resistance clusters open space for disease susceptibility [14]. In a similar way to other Southern European countries, Portugal has been subjected in the last decades to a radical neoliberalization of health care because of two major international financial crises [15]. The neoliberalization of health care has been described in the literature *vis-à-vis* healthism. Nevertheless, to the best of our knowledge, no studies have been developed in Portugal on the relation between vaccine hesitancy and healthism to date. We intend to address this gap by employing a qualitative approach and thereby to contribute to Larson and colleagues' [16] statement that the development of qualitative studies will help to enhance the understanding of vaccine hesitancy.

## 2. Background

"Vaccine hesitancy refers to the delay in acceptance or refusal of vaccines despite availability of vaccination services. Vaccine hesitancy is complex and context specific varying across time, place and vaccines. It is influenced by factors such as complacency, convenience and confidence." [2]. In addition to delay and refusal, vaccine hesitancy also refers to doubt and reluctance even by individuals who agree on vaccination regardless of their concerns. Indeed, vaccine hesitancy may occur within a broad spectrum of behaviors or beliefs [4], "from full and partial to no vaccination" [17]. There is a common understanding that the lack of trust either in healthcare professionals, governments or the pharmaceutical industry is deeply interwoven with parents' decision not to immunize their children [18–21]. The distrust on the mainstream biomedical model appears to go hand in hand with ideals of natural intensive parenting for explaining vaccine hesitancy [22–24].

The research available on vaccine hesitancy suggests that parents tend to be highly critical of mainstream biomedical fundamentals which perceived human bodies as equivalent and susceptible to diseases from biological causes [25]. In this regard, Ward and colleagues [26] proposed the term of "salutogenic parenting", referring to parents' engagement in practices they believed to naturally boost their children's immune system, thus denying the need for vaccination. Vaccine-hesitant parents usually seek more individualized and natural approaches for their children's care through healthcare practices focused on home births, extended breastfeeding, and organic and homemade food to reduce preservatives and chemicals consumption [27]. Indeed, other studies have found that vaccine-hesitant parents were concerned with the adjuvants and preservatives which vaccines may contain [28], which might contaminate children's pure bodies [29] and tend to express the desire for a "life free from chemicals and toxic ingredients" [30]. According to these authors, the view of vaccines as something artificial and dangerous was intrinsically linked to parents' fears of their potential side effects, encompassing long-term as well as

life-threatening conditions, such as mental damage, disabilities, autism, deafness, coma, cancer and even death. The child's immune system is seen as not developed, and vaccines' components are perceived to be artificial and toxic, being harmful to children's fragile bodies [31]. The framing of natural healthcare practices as safer, healthier, and less risky contributed to the deep-rooted belief that "nature offers the best way of doing things", promoting parents' commitment to natural living [32]. In addition, a shared belief that long-term immunity is better acquired through infectious diseases rather than vaccines has been found in the available literature [29]. Considering these views on childhood immunity, vaccines were framed as interfering with the natural immunity acquired from exposure to disease [30]. These parents tend to adopt an individualistic approach considering their children's singularity and, thus, reframing health as "integrated and personalized" [33].

As Wiley and colleagues [27] pointed out, "vaccination is only one aspect of parenting, and contemporary parents are subject to strong societal expectations". Within contemporary parenting culture, parents are expected to become experts in all aspects of childhood [34]. Indeed, Hays [34] describes a "model that advises mothers (and parents) to expend a tremendous amount of time, energy and money in raising their children". The salutogenic parenting practices described above appeared to be deeply interwoven with a broader social identity [27].

## 3. Methods

This article draws evidence from a wider study on the delay and refusal of childhood immunization in seven European countries (e.g., Belgium, the Czech Republic, Finland, Italy, Poland, Portugal, the UK). The VAX-TRUST project intends to understand vaccine hesitancy and improve the interaction between healthcare professionals and vaccine-hesitant parents[1]. For this article, the focus will only be on data collected in Portugal. A qualitative research approach was adopted as it offered an in-depth understanding of the experience of vaccine hesitancy, and therefore, in-depth interviews were conducted with 31 vaccine-hesitant parents between November 2021 and May 2022. These parents were recruited through snowballing (by parents or by researchers who have worked on natural intensive parenting). In addition, parents were invited to participate through an invitation via their Facebook profile (e.g., being a doula or belonging to certain Facebook communities such as those linked with "alternative" lifestyles). The boards of "alternative" schools such as Waldorf were invited to participate in the study by disseminating the project. Having a child aged 6 or under and having delayed or refused at least one recommended vaccine were the main inclusion criteria for participation. Nevertheless, our sample criteria were open to recruit parents with children above 6 years of age who have in the past refused or delayed the vaccination. We expected that this age gap would not have an impact on parents' answers regarding their children's vaccination. In fact, during the interview, all parents were very clear about the reasons for having delayed, refused, or having doubts concerning the vaccination of their children. Participants' critical reflection on vaccination began mostly during pregnancy and after their children's birth and was triggered by personal experiences such as postvaccine reactions in their children or other children and academic immunization training. Nevertheless, some parents interviewed had children over the age of 6. Prior to participation, an email was sent to parents with information about the VAX-TRUST project, and they were invited to sign an informed consent form. All interviews were conducted online via Zoom and lasted between 39 min and 1 h and 58 min. Thirty-one parents were interviewed, namely, 28 women and 3 men aged between 30 and 54 years. The majority had a university degree (*n* = 28) and, from these, four had a PhD and one was a PhD student. Most participants had a professional role in the field of health, namely, as a doula (*n* = 13), nurse (*n* = 3) and chiropractor (*n* = 3). A detailed description of the interviewed parents is provided in Table 1.

**Table 1.** Sample profile parents.

| | Personal Information | | | | Children's Information | |
|---|---|---|---|---|---|---|
| **Pseudonym** | **Sex** | **Age** | **Academic Background** | **Professional Role** | **Number of Children** | **Children's Age** |
| Mónica | Female | 38 | Degree in decorative arts | Farmer | 1 | 7 years old |
| Manuel | Male | 45 | Master's Degree | Musician; Teacher | 4 | 18, 18, 11 and 2.5 years old |
| Maria | Female | 33 | Degree in Nursing | Nurse and Naturopath | 1 | 2 years old |
| Madalena | Female | 36 | PhD Student | Researcher | 1 | 6 years old |
| Mariana | Female | 40 | PhD | Social Psychologist; Researcher | 2 | 11 years old; 1 year and 10 months old |
| Mafalda | Female | 43 | Postgraduate | Psychopedagogue | 2 | 14 and 7 years old |
| Márcia | Female | 40 | Degree; (Master's Degree student) | Anthropologist | 1 | 9 years old |
| Miguel | Male | 54 | Degree in Agricultural Engineering | Technical Director of a Water and Environment Company | 4 | 20, 19, 5 years old and 1 month old |
| Margarida | Female | 37 | Degree in Nursing | Farmer and information analyst | 1 | 5 years old |
| Magda | Female | 41 | Degree in Physical Education | Businesswoman; Chiropractic student | 1 | 2 years and 4 months old |
| Marina | Female | 44 | Master´s Degree in Health and Sports/Exercise | Doula; Personal Trainer | 2 | 17 and 15 years old |
| Melissa | Female | 44 | Master's Degree | Chiropractor (specialized in pregnant women and children) | 4 | 16, 14, 13 and 10 years old |
| Marta | Female | 30 | Master's Degree | Chiropractor | 2 | 6 and 5 years old |
| Marlene | Female | 40 | Degree in Physiotherapy; postgraduate in Pediatric Physiotherapy | Doula; Physiotherapist | 1 | 5 years old |
| Nádia | Female | 41 | Degree and postgraduate in immunology | Nutritionist | 1 | 6 years old |
| Natália | Female | 38 | Degree in Psychology | Doula; Master's Degree student in Clinical Psychology | 2 | 10 and 7 years old |
| Mario | Male | 43 | PhD | Management consultant | 2 | 8 and 7 years old |
| Natacha | Female | 36 | Degree in Nursing; Master's Degree in Health Sociology; PhD in Sociology | Former nurse; Professor and Researcher at the Faculty of Medicine | 2 | 5 and 2 years old |
| Noémia | Female | 39 | Degree in Design | Designer | 2 | 4 years and 3 months |
| Núria | Female | 37 | Degree in Archaeology and Piano | Scientific Research Grant Holder FCT—in Music and History | 2 | 9 years and 15 months |
| Nazaré | Female | 33 | Degree in Education | Doula | 1 | 4 years old |
| Neide | Female | 37 | Secondary school | Doula | 3 | 17, 13 and 3 years old |
| Nicole | Female | 31 | Degree in Engineering | Quality control in her family business; Doula | 2 | 4 and 2 years |
| Neuza | Female | 33 | Degree in Nursing | Doula | 1 | 2 years and 5 months |
| Naomi | Female | 44 | Secondary school | Yoga and Pilates teacher; Doula | 3 | 13, 11 and 8 years old |
| Nadine | Female | 46 | Degree in Architecture | Doula | 4 | 18, 16, 14 and 10 years old |
| Lara | Female | 41 | Degree in Nursing | Nurse in a health centre; Doula | 2 | 11 and 2 years and a half |
| Laura | Female | 39 | Secondary school | Doula | 3 | 17, 15 and 3 years and 5 months |
| Lisa | Female | 35 | Degree in Public Relations | Doula | 3 | 8, 7 and 4 years old |
| Leonor | Female | 46 | PhD in Educational Science | Natural Science Teacher and Women's Health Therapist | 1 | 13 years old |
| Letícia | Female | 45 | Degree in Nursing | Doula; Independent specialist nurse in maternal health | 3 | 9, 5 and 4 years old |

All interviews were conducted in Portuguese and recorded with participant permission and transcribed verbatim. The interviews were analyzed using NVivo 1.6.1 version. Ethical approval was obtained from the Instituto de Ciências Sociais, Universidade de Lisboa—host research center of the VAX-TRUST project in Portugal. A thematic analysis was developed [33]. After an open-coding procedure, first-order themes were reduced into more restricted themes. Then, after a deep reading, these restricted themes were reduced to core themes. Thereby, six themes were identified: (i) natural birth; (ii) extended breastfeeding; (iii) adoption of a vegetarian/macrobiotic diet; (iv) preference for alternative educational models; (v) "natural medicine"; (vi) distrust in science. These themes will be discussed in the next section. Illustrative quotations translated to English will be presented.

## 4. Results

### 4.1. Natural Birth

Hesitant parents mentioned that they tried to have their child's births be as natural as possible. In most cases, parents decided to have a home birth with the support of qualified professionals such as a doctor, a nurse–midwife and/or a doula. Alternatively, parents who chose to have their babies in a hospital setting sought to ensure that the birth took place in the most natural and humanized way, even if that implied an increased suffering of the parturient ("I had a normal delivery without epidural, but it was my choice. And with as little medication as possible.", Naomi). This was also the case of Magda, who opted for having her baby in a private clinic due to not having the necessary conditions for a home birth:

> "[The clinic] is known for providing the most natural birth possible (. . .) It was highly recommended for those who wanted to have a non-medicated birth. So, basically, I went to the clinic a little bit against my will because at the time, I was trying to have a home birth. What happened was that, at the time, we were living in a rented house, and it did not have the conditions that my husband thought were ideal for the baby to be born at home." Magda (41, one child)

Even in cases where deliveries took place in a hospital, hesitant parents tended to devalue the role of the obstetrician and, in contrast, highlighted women's capacity and self-sufficiency to give birth autonomously. For instance, Nádia indicated:

> "I think 90, over 95% of pregnancies, the obstetrician is just there to see the thing, right? The woman can do the job, she just needs the support but there is that 5% that makes the difference. So, I think the obstetrician's role is really 'ok, I am here just in case' (laughs) is it not? And if anything happens, they are there to intervene and intervene well." Nádia (41, one child)

In addition, Marina, an interviewed mother who was also a doula, stressed that, according to her work experience, there is a correlation between parents who strive for a natural birth and vaccine hesitancy, "And the truth is that people who opt for home birth are people who question a lot of things and one of the things they question a lot is vaccination." (Marina, 44, two children).

### 4.2. Extended Breastfeeding

Hesitant parents pointed out extended breastfeeding as a key strategy to promote the development of their children's immune system. More specifically, they advocated breastfeeding on free demand based on the assumption that it allowed children to become naturally immunized and, consequently, more robust to deal with any illnesses. In some cases, hesitant parents highlighted that their decision to breastfeed their children was considered and prepared before the child's birth, having been set as a family goal:

> "We had already thought we had the intention of breastfeeding (. . .) It was mobilising resources so that I would have support regarding this aspect, right? Looking for people who could support me to achieve our goal as a family." Marlene (40, one child)

Parents' discourses revealed that this process did not always run smoothly, requiring parental effort, especially from mothers, who relied on a support network to overcome breastfeeding-related difficulties, as illustrated in Noémia's discourse, "At first, breastfeeding did not go very well with the first child. I made an effort. I had a breastfeeding counsellor coming to my home to start and manage breastfeeding. . ." (Noémia, 39, two children).

The interviewed mothers' sense of commitment to breastfeeding was well illustrated in the discourse of Laura, a mother of three children, who self-recriminates for having stopped breastfeeding her oldest daughter at "just" one year of age, "I even went to sleep last night thinking: 'So absurd! Completely disrespectful.' For three years now, until today, I suffer because [at the time] I thought her weaning was a success." (Laura, 41, three children). As a way of redeeming herself, this mother mentioned that she was presently still breastfeeding her youngest daughter, aged three and a half years.

As hesitant parents shared the perception of breastfeeding as a crucial booster of children's immune system, they downplayed the relevance of vaccination, relying on children's natural immunity over an artificial one. This argument was used by hesitant parents to justify their decision to postpone or even reject vaccination along with other factors such as children's environment, "And as long as there was the issue of breastfeeding, and the children were not in contact with other children, we decided that there would not be no kind of vaccination." (Natália, 38, two children).

### 4.3. Adoption of a Vegetarian/Macrobiotic Diet

Hesitant parents advocated a healthy and varied diet for their children as a key strategy to promote their health and prevent diseases, especially during the first years of life, "Basically, my health care for Augusto is to make sure he eats well, right? Have the foundations he can have so he does not get sick." (Magda, 41, one child). In most cases, hesitant parents showed a greater preference for a vegetarian or macrobiotic diet, prioritizing vegetables, fruits and fish over meat, "Our diet is also very focused on vegetables, fungi, mushrooms and essentially nuts, all those kind of things. . . nuts, oilseeds, pulses. . ." (Nazaré, 33, one child).

When reasoning about their diet choices, hesitant parents pointed out their children's health as well as environmental reasons, prioritizing organic instead of processed foods such as refined flour or sugar, ". . . the child does not eat sugar, especially during the first years of life, while we make sure that his immune system is as strong as possible." (Maria, 33, one child). The adoption of an organic diet required parental efforts such as those illustrated in the discourse of a mother of a 4-year-old boy who bakes bread at home, ". . .It is always a difficult balance (. . .) We try as much as possible to cook things at home when we have time for it." (Nazaré, 33, one child).

Despite parents' strong beliefs about the most appropriate diet for their children, they showed flexibility in its implementation by advocating, for instance, a reduction in meat consumption rather than its definitive elimination from their children's diet. In addition, hesitant parents highlighted the need to consider personal characteristics such as individual nutritional needs (for instance, the ones deriving from the low meat intake) when adopting a vegan or vegetarian diet, as shown in Mafalda's discourse about the management of her two daughters' diet:

> "With Camila we look at her needs, and with Aurora we look at Aurora's needs. It is by looking at the needs they have. . . And the stage of development they are in, that we adjust the diet. And that is why I say once again that it is a functional diet." Mafalda (43, two children)

In some cases, parents relied on medical supervision to ensure that their children's nutritional needs were met:

> "As soon as we started introducing food, we had a nutritionist who accompanied us until he was 12 months old (. . .) With a child it is necessary to take a different

kind of care and we wanted to know if we were doing things carefully and supplying all his nutritional needs ..." Neuza (33, one child)

Some interviewed parents living in the countryside highlighted their easy access to organic food which they produce themselves or, alternatively, is produced by neighbours in their vegetable gardens. Conversely, hesitant parents living in an urban context reported difficulties in following a rich and healthy diet which, along with other consequences of urban living, led to the need for vaccination as explained by Márcia:

"... given the nutrient deficient foods we eat, lack of contact with pure environments, excessive pollution, overpopulated places where there is more disease contagion, the only way such a society, with the existing level of world population can function, is with vaccination." Márcia (40, one child)

### 4.4. Preference for Alternative Educational Models

When asked about their choices on children's education, hesitant parents showed a clear preference for alternative educational models such as the modern school movement, Montessori, Waldorf, Reggio Emilia and the Forest School approaches. Despite the individual characteristics of each of these alternative models, they share some core values such as the promotion of children's contact with nature, the advocacy of a more individualized and less-massified education, a greater respect for the personal characteristics and preferences of each child and fostering self-empowerment, as was shown in a mother's discourse about the Montessori pedagogy:

"The respect for the individual, the respect for nature and the integration of nature into the routines of Anselmo's living. This is a methodology of much observation of the child, of... Many rituals. These are things that I also value, as a person, for me." Marlene (40, one child)

Additionally, the interviewed parents outlined the school diet offer as a distinguishing factor, being in line with the one at home, "He has this biological diet, right? Balanced, so, he... So, for me, at school, this is a very important weapon for him to have..." (Margarida, 37, one child).

In some cases, parents mentioned making great efforts for their children to attend this type of school such as walking long distances every day or even choosing the place to live based on the location of the school, as shown in Margarida's discourse, "Among the Waldorf schools, we chose one that we wanted him to go and then we moved [laughs] to live where it was close to the school" (Margarida, 37, one child).

However, in some cases, hesitant parents mentioned that, even though they wish their children to attend a school with an alternative model, they were not able to do so due to location or financial-related constraints. Regarding the latter, parents criticized the unaffordable prices of alternative private schools which were only reachable by the most privileged, as was illustrated in a mother's statement describing the attendance of such schools as similar to "living in a bubble": "Although I am aware that I would prefer my son to go to an alternative school, they are all very expensive (...) As they are very expensive, they are only for some people." (Natacha, 36, two children).

As an alternative, parents searched for public schools offering a different program from the conventional by including, for example, artistic education, or being located in more rural areas (e.g., "in a farm"), allowing children to have a broader space to play as well as greater contact with nature:

"... the school is also more in the countryside and it is a school with a fairly large outdoor space (...) One of the main characteristics of that school is that children are very involved in the garden, they have a greenhouse and produce many things and help with the organic fertilizer, and go weeding and sowing and then the school also produces food hampers." Madalena (36, one child)

In addition, the opportunity to interact with animals was another determining factor in choosing a school for their children:

"... it is a farm. My children love animals, bushland, farms, horses, they are very used to the outdoors and the school is a college, but outside is a huge farm, it is not a space between walls [. . .] He has contact with animals: cow, chicken, dog, peacock. That was the key deciding factor." Nicole (31, two children)

However, in some cases, hesitant parents enrolled their children in conventional public schools due to a lack of alternatives, being sharply critical about its underlying ideology, as illustrated in Neide's discourse:

"(. . .) [public schools] have a pedagogical method and it is that pedagogical method for everyone. This is completely wrong. Teaching should be adapted to each child because each child is unique and special. And they are often very intelligent children, but they cannot adapt to that teacher's methodology. And they are labelled as stupid (. . .) because they cannot understand the way the teacher is teaching them something." Neide (37, three children)

Beyond the preference for schools with alternative educational models, hesitant parents also mentioned postponing the enrolment of their children in school while maximizing the time they stay at home until the age of three or four. The extension of the time children remained at home in their first years of life, along with other natural immunity-boosting strategies, were used to minimize their exposure to eventual diseases, "Vitamin D supplementation, microbiotic supplementation, breastfeeding, healthy diet. (. . .) Not having gone to school, so that way he was also protected." (Nádia, 41, one child).

In a few cases, hesitant parents advocated for homeschooling as an alternative for their children, despite it being a complex process due to administrative requirements. In this regard, Mafalda made an analogy between homeschooling and vaccination, arguing that there is a gap between the law and the practice:

"She went to home-schooling which is, like vaccination, a freedom of choice that we have in Portugal. It is not often recognised as, nowadays, it is almost false (. . .) So the law provides that parents have the right to choose the type of education they want to give to their children. At this stage, if we were to put our child in home-schooling, we would have to give in a legitimate document stating that this kind of teaching exists, together with a whole lot of other things, it´s rather a megalomaniac process." Mafalda (43, two children)

Based on our findings, hesitant parents' choices on their children's education did not occur isolated but, rather, as part of their lifestyle practices and health beliefs, "He goes to a Waldorf school, does he not? So, we choose to go for an education that is in line with our philosophy, our lifestyle and yet has the basis of anthroposophical medicine. . ." (Margarida, 37, one child).

### 4.5. Natural Medicine

Hesitant parents only considered giving medication to their children when they found it to be strictly necessary. The avoidance of medicalization was especially true regarding antibiotics, which were perceived as the "cutting-edge solution", as illustrated in Nazaré's discourse:

"We are aware and we have confidence in the paediatrician that when we have tried everything and nothing is working, if the antibiotic is needed, the antibiotic is needed" (Nazaré, 33, one child)

Hesitant parents adopted a "natural medicine" based on the consumption of vitamins, minerals, magnesium, iron, propolis and echinacea, as they believed that this would improve their child's immune system. Indeed, these supplements were generally given to children prophylactically and in a planned way, especially before and during the winter period, when diseases are most likely to occur, as illustrated in Mafalda's discourse:

"We always do prevention (. . .) it is a planning (. . .). Planning at a nutritional level, so we do supplementation, in order to prevent exactly that. There are periods of the year when we invest, like this one, in the immune system. And we do supplementation, vitamin C, vitamin D, zincs, all minerals, all that type." Mafalda (43, two children)

In the event of their children becoming ill, some hesitant parents also relied on home remedies, even if that implied more time to recover and an increased parental monitoring effort. As an example, Nicole described her choices as a result of her daughter's urinary tract infection:

"Using the natural method, watching for three days and seeing if it improves. (. . .) I did not give her antibiotics. (. . .) Then I went on to do a rosemary bath, citrine salt which is anti-sceptic. . .". Nicole (31, two children)

She added that the use of this "natural medicine" as a strategy to manage children's health also extended to the family, "It is the same thing with headaches, in our house nobody takes ben-u-ron or paracetamol, we smell mint. (. . .) we make chamomile (. . .) And the essential oils too, of course." (Nicole, 31, two children).

In a nutshell, hesitant parents' choices to manage their children's health were in line with their commitment to natural living, denoting a distrust in conventional medicine. Thereby, showing healthist attitudes.

*4.6. Distrust in Science*

What also became evident in our findings was parents' distrust in science and medicine. Most interviewees spontaneously mentioned the hepatitis B vaccine as the triggering point of their doubts, as this vaccine is usually given to newborns still in the maternity ward. For instance, a mother stated:

"Why am I already going to interfere with this kid's immune system when he is one or two days old to protect him from a disease that his mum might not even have, what is the likelihood of him getting it?" (Nádia, 41, one child)

Hesitant parents' arguments to postpone or refuse the hepatitis B vaccine usually relied on its perception as unnecessary, as children do not engage in risky behaviours such as sexual intercourse or sharing of needles. This was expressed by Natália, who acknowledged that their children may later choose to be vaccinated or not:

"And then the hepatitis B [vaccine]. . . Which will be their choice when they get older, when they start having sex with other people. Then we have to rethink this vaccine. I feel that they will already be able to participate in the decision and to make their choice." (Natália, 38, two children)

Throughout childhood, parents continuously decide about vaccinating their children on a case-by-case basis considering both the severity and the prevalence of each disease. Regarding the former, some hesitant parents mentioned the MMR vaccine as needless by sharing the view of measles as a common disease in childhood and minimizing its potential health risks:

"I think that measles is a disease that anyone with a good, a strong immune system can cope with, I don't know if there is a need for all the hysteria about measles." (Márcia, 40, 1 child)

Regarding the prevalence of each disease, polio was mentioned as an example of a disease considered to be "eradicated", and thus, with no need for immunization:

"[The polio vaccine] it is compulsory and the disease is totally eradicated. In other words, there has been no case of polio in Portugal or in other countries for some years now. Therefore, it is assumed that it is eradicated. (. . .) it doesn't make much sense to have a vaccine now." (Leonor, 46, 1 child)

These quotations above are illustrative of misconceptions of vaccine-hesitant parents regarding vaccination and the prevalence of vaccine-preventable diseases. This was augment by the lack of effective communication between parents and healthcare professionals. Indeed, healthcare professionals generally reply evasively, and according to our participants, they used derogatory language to blame and pressure vaccination. This was the case of Maria, who indicated that her decision to postpone the hepatitis B vaccine was heavily criticized by healthcare professionals:

> "The conversation arises from why was this not done, do you have any idea, do you realise how dangerous this is . . . and how are we going to do it now? Are you not going to do it? The question is always this. Are you not going to do it? You are not going to vaccinate, is that it? And I never experienced the other side which is. . .let's talk about it." (Maria, 33, one child)

## 5. Discussion

The findings of this study show that healthism beliefs are likely to go hand in hand with vaccine hesitancy among Portuguese parents. This is in accord with the work of Swaney and Burns [35], who found that vaccine-hesitant parents often believe that natural lifestyle choices are key to control vaccine-preventable diseases. Indeed, when reasoning about their choices regarding their children's health, the arguments of the hesitant parents interviewed for the current study relied on the dichotomy of natural and artificial [14], favoring the first. Their focus on natural living made them prioritize a natural birth with little to no medication, highlighting woman's autonomy and their bodies' natural ability to perform labor without any kind of intervention. Alternatively, they resorted to natural supplements as they believed that this would boost their children's immunity in a prophylactic way, aiming to minimize the impact of an eventual illness. In the same vein, when children got sick, hesitant parents relied on home remedies, which implied close parental supervision. For these parents, the choice of natural products goes hand in hand with the idea of being less risky [36].

In addition to natural birth, hesitant parents stressed the relevance of extended breast-feeding as the most natural and, thus, most effective means to promote babies' and children's immunity. This is in line with the assumption that "nature offers the best way of doing things", making breastfeeding perceived as safer, healthier and less risky than vaccination [32]. The deep-rooted belief about the relevance of breastfeeding triggered mothers' commitment and sense of responsibility to this cause, making them feel accountable for their children's natural immunity. Hesitant parents continued to manage their children's nutritional intake throughout childhood by feeding them organic homegrown food and prioritizing a vegetarian or macrobiotic diet [37]. Moreover, they preferred cooking from scratch rather than using processed foods containing additives and preservatives, even if that implied increased parental effort.

Hesitant parents' natural living extended to their views about their children's education by choosing schools providing alternative educational models. This option was very much related with parents' holistic approach to raising children [38]. Indeed, hesitant parents were highly critical of conventional public schools which "place children between four walls" while applying a single pedagogical method without accounting for children's individual characteristics. The connection between vaccine hesitancy and preference for alternative pedagogical models, rather than a massified education, had already been found in previous studies conducted in the U.S. [38] and in Australia [26]. To date, however, this relation has not been sufficiently described in the European literature. Moreover, in some cases, hesitant parents mentioned postponing their children's enrolment in kindergarten as a strategy to protect them against infection during their first years of life. Parents' control of their children's social networks by extending the period they stayed at home was also found in a study with American mothers [39]. Previous research on vaccine hesitancy suggested that practices around vaccination are deeply interwoven with parents' experiences regarding a child's health and institutions [40].

Based on our findings, hesitant parents' views on their children's health were in line with the concept of salutogenic parenting as converging the health-promoting and illness-preventing practices mentioned above [26]. The interplay of the caregiving practices described above was believed to ensure children's health in a natural way and making vaccines needless or, at least, less necessary [27]. This view embodied the concept of healthism as shedding light on the individual responsibility and control for health management [14]. Indeed, parents understood themselves as "active and capable agents" who have the skills and the knowledge to make the best decisions for their children [26]. Hesitancy towards vaccination may be understood as a result of the focus of health promotion and individual agency [41]. Vaccine-hesitant parents recognize themselves as the main "experts" on their children's health, relying more on their own judgment to weigh the pros and cons of vaccination rather than on healthcare providers' recommendations [39]. Parents understood themselves as guardians of children's pure bodies, which need to be protected from contamination by outside sources [42,43]. Healthist attitudes are illustrated throughout the criticism of these parents towards "unnatural substances" such as vaccines, favoring the natural immunization of their children.

The demands of intensive parenting [44] were similarly experienced by mothers and fathers interviewed for our study. Nevertheless, our sample was mainly composed of mothers. Previous research by Reich (2014) found that mothers recognize themselves as the main "experts" on their children's health, relying more on their own judgment to weigh the pros and cons of vaccination rather than on healthcare providers' recommendations [39]. There is evidence that healthcare decisions are mostly taken by mothers [45] and, accordingly, vaccine hesitancy may be considered as a gendered process, given women's central role in assessing the benefits and risks of the vaccination for their own children [46]. Nevertheless, in our findings we did not find gender differences, as both mothers and fathers expressed concerns about vaccinating their children. What clearly became evident was that the parents interviewed for our study assume responsibility for the health of their children and manage their health risks through the adoption of healthist attitudes which are aligned with the neoliberalism ideology.

Most hesitant parents in our sample spontaneously pointed out the hepatitis B vaccine as the triggering point of their doubts about vaccination, as they considered it as being needless in newborns. This position stands on their beliefs about the routes of disease transmission and on their perceived lack of evidence offered by healthcare professionals. These parents' distrust in science and medicine is very much aligned with the ideals advocated by healthism. Indeed, the two dimensions of healthism (personal responsibility for their own health and distrust in medical authorities) are made apparent in the discourses of vaccine-hesitant parents [47]. This commitment to healthism and its association with vaccine hesitancy has been found to be very much present in educated middle classes [48,49].

However, the choices made by vaccine-hesitant parents represented a privileged position, as the pursuit of healthfulness is constrained by sociodemographic aspects [49]. The commitment with children's health implied ongoing parental supervision requiring additional time and money: extended breastfeeding, exhaustive searching for information about home/natural births (e.g., identification of healthcare professionals with similar ideals), tailoring diets to each child's nutritional needs, home remedies which usually implied more time to recover from illness, and preference for private schools with alternative educational models which are both more expensive and, in some cases, farthest from home. These practices were very much related with the fact that the parents in our sample were generally middle class; in particular, they were highly educated and had a good financial position. This resonates with the findings of Ward and colleagues [26]. Indeed, as outlined by Reich [39], the decision of delaying or refusing the vaccination of their children may be middle-class privileged. Vaccine hesitancy may to a certain extent reflect social inequality, as those who are more well educated and have certain financial resources are more able to choose [39,50,51]. Unlike previous studies that found a broad spectrum [27], and even

though most of the participants in our study were not connected with one other, we noted a common approach among vaccine-hesitant parents.

Throughout their discourses, hesitant parents revealed to be very proactive and felt empowered searching and choosing the "best options" to manage their children's health, which appears to be aligned with the concept of health literacy[2] [52]. Although we did not measure participants' health literacy level in the present study, we would expect it to be high given that most of them had tertiary education, and the literature shows that there is a positive correlation between health literacy and educational attainment [53]. There are mixed findings in the literature regarding the relation between health literacy and vaccine hesitancy. In a recent systematic review, Lorini and colleagues [54] found this relation to be influenced by factors such as risk perception and the likelihood of contracting the disease and its short-term consequences. On one hand, when these probabilities were perceived as high, health literacy positively predicted vaccination uptake; on the other hand, when these were low, a negative relation between health literacy and vaccination uptake was found. These conclusions may possibly shed light on the findings of the current study, as hesitant parents estimated a very low chance of their children contracting certain vaccine-preventable diseases. Based on our findings, we suggest that future studies should further explore the influence of both health literacy and communication between parents and healthcare professionals on parental decisions on children's immunization.

### 6. Conclusions

We argue that the delay or refusal of vaccination should also be understood as a pursuit of healthiness by vaccine-hesitant parents. Healthism draws upon the assumption that the burden of health care should be placed on the shoulders of individuals and that this would be beneficial not only for individuals themselves but to society as a whole [8]. However, using vaccine hesitancy as the starting point, our findings show that healthism and its focus on personal accountability under the umbrella of neoliberalism may jeopardize global public health. While vaccine-hesitant parents in our sample focused on the needs of their own children, they may have put the health of others at risk by not following mainstream public health policies regarding vaccination [39]. To address the distrust in science and medicine evident in the discourse of vaccine-hesitant parents, healthcare professionals should spend more time discussing parental vaccination concerns. We propose that the presumptive or paternalistic model of communication, which has proved ineffective in cases of vaccine hesitancy as information is usually perceived as biased [55], should be replaced by a patient-centered approach, which takes individual/family characteristics and preferences as a starting point. Indeed, the findings of this study recall that healthcare professionals should pay particular attention to healthism when addressing vaccine hesitancy. Research evidence advocates the need to be sensitive to the broad spectrum of vaccine hesitancy, as this encompasses multiple views on the subject [56,57]. Indeed, the labeling of "anti-vaxxers" has been found to increase the polarization of discourses on vaccination, having a backfire effect in vaccine acceptance [58,59]. Therefore, instead of being critical of vaccine-hesitant parents, a more comprehensive approach on the reasons underlying vaccine hesitancy should be developed [59].

### 7. Study Limitations

This research has some limitations. The first is related to the lack of heterogeneity. All the parents interviewed were white. Future research should also focus on migrant parents or parents from ethnic minorities, as there is evidence from previous studies that pockets of low immunization tend to occur in these populations [60]. The parents interviewed for this study were middle-class, confirming previous findings from other work [39]. Therefore, other studies should include parents from other social positions. In addition, other sociodemographic aspects such as the age of the parents and the child should also be explored. Further research should examine gender differences in detail among parents regarding vaccine hesitancy, because although most studies have focused on mothers,

fathers may also have a word to say concerning the vaccination of their children. The small sample of parents (*n* = 31) means that generalizability cannot be made to vaccine-hesitant parents in Portugal or elsewhere.

**Author Contributions:** Conceptualization: A.P.H. and J.M. Methodology: J.M. and A.P.H. Formal analysis: Joana Mendonça. Writing: J.M. and A.P.H. Funding acquisition: A.P.H. All authors have read and agreed to the published version of the manuscript.

**Funding:** This project has received funding from the European Union's Horizon 2020 research and innovation programme under Grant Agreement N.º 965280.

**Institutional Review Board Statement:** Ethical approval was obtained from the Instituto de Ciências Sociais da Universidade de Lisboa (Ref: 2021_16).

**Informed Consent Statement:** Written informed consent has been obtained from the parents to publish this paper.

**Data Availability Statement:** The data presented in this study are available on request from the corresponding author. The data are not publicly available due to the sensitivity of the theme under study.

**Conflicts of Interest:** There are no conflicts of interest to declare.

## Notes

1   For further information on the methodology of the project please see: Cardano, M.; Numerato, D.; Gariglio, L.; Marhánková, J.; Scavarda, A.; Bracke, P.; Hilário, A.P.; Polak, P.; Hobson-West, P.; Vuolanto, P. A rapid team ethnography on vaccine hesitancy in Europe: methodological reflections, *under review*.

2   Health literacy is defined by the World Health Organization as representing "the cognitive and social skills which determine the motivation and ability of individuals to gain access to, understand and use information in ways which promote and maintain good health" [52] (p. 357).

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
