# Peer review of "Healthism vis-à-vis Vaccine Hesitancy: Insights from Parents Who Either Delay or Refuse Children’s Vaccination in Portugal"

_societies, doi:10.3390/soc13080184_

Round 1

Reviewer 1 Report

The paper deals with healthism and vaccine hesitancy among middle-class parents who delay or refuse children’s vaccination in Portugal.

The topic is highly relevant for various fields, including public health.

However, there are several questions I would like to raise about the paper.

1.

»Healthism draws from neoliberal discourse that highlights personal responsibility and the need for self-management.«

Neoliberalism is mentioned several times concerning healthism, yet it should be more clearly noted how they are linked (i.e. one emerges within the other).

2.

The biggest drawback of the paper is a lack of studies cited that examined the healthism and vaccine hesitancy link in the past. Since very few exist on this link, I would suggest finding them and citing them to provide context to the reader in either the Introduction or Discussion section. The authors may want to use MDPI journals and other publications on healthism and vaccine attitudes and hesitancy. At least four examples of papers can be found in the literature:

1. https://doi.org/10.3390/su15076107

2. Swaney, S.E.; Burns, S. Exploring reasons for vaccine-hesitancy among higher-SES parents in Perth, Western Australia. Health Promot. J. Aust. 2019, 30, 143–152.

3. Peretti-Watel, P.; Raude, J.; Sagaon-Teyssier, L.; Constant, A.; Verger, P.; Beck, F. Attitudes toward vaccination and the H1N1 vaccine: Poor people’s unfounded fears or legitimate concerns of the elite? Soc. Sci. Med. 2014, 109, 10–18.

4. Bocquier, A.; Fressard, L.; Cortaredona, S.; Zaytseva, A.; Ward, J.; Gautier, A.; Peretti-Watel, P.; Verger, P. Social differentiation of vaccine hesitancy among French parents and the mediating role of trust and commitment to health: A nationwide cross-sectional study. Vaccine 2018, 36, 7666–7673.

3.

All interviews were conducted online via zoom and lasted between 00:39 and 1:58 minutes..

I suggest rewriting in a more transparent manner: 39 minutes and 1 hour and 58 minutes.

4.

“Thirtyone  parents  were  interviewed,  namely  28  women  and  3  men  aged  between  30  and  54 years.”

It should be discussed how the gender imbalance of the sample impacted the findings.

In addition, do men in the sample (although only three) differ with regard to healthism and vaccine hesitancy?

5.

The Results are basically stating five main themes with illustration quotes. This is not methodologically sound/detailed enough, in my opinion, especially compared to other qualitative studies on vaccine hesitancy.

6.

Results essentially discuss the themes but sufficiently not the most central theme – the link between healthism and vaccine hesitancy. This should be the most stressed topic in the Results section.

7.

“The neoliberal discourse within which is at the heart of healthism ideology emphasis individual lifestyle choices (41)”

This is an example of a sentence that is not understandable to the reader.

Another erroneous one:

“Further research should also examine gender differences amongst parents regarding vaccine hesitancy, as even though most studies have focused on mothers, parents may also have a word to say concerning the vaccination of their children.”

Are mothers not regarded as parents?

8.

The Study limitations section needs to be expanded.

9.

Policy recommendations should be more concrete and informative to policymakers.

10.

Considering the above-stated comments and open questions, I advise to revise and resubmit.

Please, see above.

Reviewer 2 Report

Thank you for the opportunity to review this very interesting paper.

It would be useful to state what language the interviews were conducted in, was it Portuguese or English. If English did this further reduce the respondents able to participate. If the questions and answers were translated, some discussion on some of the possible translation of terms or ideas faced by the translator and then the research team is required.

I think the work would be improved if it included a more comprehensive description of the study respondents. The study has 31 respondents gained from a snowballing recruitment process; I’m not sure that this group can be classed as representative of the wider vaccine hesitant parents in relation to healthism as seen in Portugal, as alluded to in the paper. Looking at table 1 it appears that 9 respondents (almost a third of the total group) have children over the age of 6 years, the age considered optimal by the researchers. This factor is not discussed in any depth and may have had an impact on parents’ memories of reasons for none vaccine attendance.  It would also be helpful for the reader to understand which vaccines the respondents refused, reasons for refusing hpv, or mmr vaccine are known to differ from the reasons offered for the initial vaccinations given in the first year of life. The respondents in this study self-selected. Consequently, information on economic status, ethnicity, geographical location, health and wellbeing of children and what sources the parents accessed to make their vaccine decisions. As almost a third of respondents had their children some time ago what trusted sources were available when these children were babies and have trusted sources improved or declined in Portugal during this time. This would be useful to help formulate strategies to support such families.

Some discussion of these findings in relation to health literacy would also be valuable. People with higher educational attainment are generally thought to have higher health literacy, while health literacy is concerned with the abilities people require to access, appraise, understand and use information to enhance health. Could it be that although this group of people have higher educational qualifications and may well be empowered and enabled, their health literacy around vaccines for children is low and as individuals they have a lack of concern for the wider community contradicting the WHO’s work on the topic, which maintains that health literacy in populations enables improvements in personal health and acts collectively to lobby governments to meet their responsibilities in addressing health and equity.

Overall the paper raises some interesting themes. 

Overall the paper raises some interesting themes. Very minor spelling and sentence construction issues.

Round 2

Reviewer 1 Report

The paper is now improved. I suggest the authors check the references and publication years, as there are some typos (wrong years stated etc.).

Please, see above.

Author Response

Response to Reviewer 1 Comments

Point 1: The paper is now improved. I suggest the authors check the references and publication years, as there are some typos (wrong years stated etc.).

Response: 

All the references were checked and the existing typos were corrected. One reference was duplicated (49) and, consequently, all the following references were renumbered.    Thank you very much for drawing our attention to this matter.